# Regulatory T Cell Apoptosis during Preeclampsia May Be Prevented by Gal-2

**DOI:** 10.3390/ijms23031880

**Published:** 2022-02-07

**Authors:** Sarah Meister, Laura Hahn, Susanne Beyer, Mareike Mannewitz, Carolin Perleberg, Konstantin Schnell, David Anz, Stefanie Corradini, Elisa Schmoeckel, Doris Mayr, Uwe Hasbargen, Alaleh Zati Zehni, Sven Mahner, Udo Jeschke, Thomas Kolben

**Affiliations:** 1Department of Gynecology and Obstetrics, University Hospital, LMU Munich, Marchioninistr. 15, 81377 Munich, Germany; sarah.meister@med.uni-muenchen.de (S.M.); laura.hahn@med.uni-muenchen.de (L.H.); susanne.beyer@med.uni-muenchen.de (S.B.); mareike.mannewitz@med.uni-muenchen.de (M.M.); uwe.hasbargen@med.uni-muenchen.de (U.H.); alaleh.zati@med.uni-muenchen.de (A.Z.Z.); sven.mahner@med.uni-muenchen.de (S.M.); thomas.kolben@med.uni-muenchen.de (T.K.); 2Center of Integrated Protein Science Munich (CIPS-M), Division of Clinical Pharmacology, University Hospital, LMU Munich, 81377 Munich, Germany; carolin.perleberg@med.uni-muenchen.de (C.P.); konstantin.schnell@gmail.com (K.S.); david.anz@med.uni-muenchen.de (D.A.); 3Department of Radiation Oncology, University Hospital, LMU Munich, Marchioninistr. 15, 81377 Munich, Germany; stefanie.corradini@med.uni-muenchen.de; 4Department of Pathology, University Hospital, LMU Munich, Marchioninistr. 15, 81377 Munich, Germany; elisa.schmoeckel@med.uni-muenchen.de (E.S.); doris.mayr@med.uni-muenchen.de (D.M.); 5Department of Gynecology and Obstetrics, University Hospital Augsburg, 86156 Augsburg, Germany

**Keywords:** regulatory T cells, apoptosis, preeclampsia, Galectin-2

## Abstract

There are several open questions to be answered regarding the pathophysiology of the development of preeclampsia (PE). Numerous factors are involved in its genesis, such as defective placentation, vascular impairment, and an altered immune response. The activation of the adaptive and innate immune system represents an immunologic, particularity during PE. Proinflammatory cytokines are predominantly produced, whereas immune regulatory and immune suppressive factors are diminished in PE. In the present study, we focused on the recruitment of regulatory T cells (Tregs) which are key players in processes mediating immune tolerance. To identify Tregs in the decidua, an immunohistochemical staining of FoxP3 of 32 PE and 34 control placentas was performed. A clearly reduced number of FoxP3-positive cells in the decidua of preeclamptic women could be shown in our analysis (*p* = 0.036). Furthermore, CCL22, a well-known Treg chemoattractant, was immunohistochemically evaluated. Interestingly, CCL22 expression was increased at the maternal-fetal interface in PE-affected pregnancies (*p_syncytiotrophoblas_*_t_ = 0.035, *p_decidu_*_a_ = 0.004). Therefore, the hypothesis that Tregs undergo apoptosis at the materno-fetal interface during PE was generated, and verified by FoxP3/TUNEL (TdT-mediated dUTP-biotin nick end labeling) staining. Galectin-2 (Gal-2), a member of the family of carbohydrate-binding proteins, which is known to be downregulated during PE, seems to play a pivotal role in T cell apoptosis. By performing a cell culture experiment with isolated Tregs, we could identify Gal-2 as a factor that seems to prevent the apoptosis of Tregs. Our findings point to a cascade of apoptosis of Tregs at the materno-fetal interface during PE. Gal-2 might be a potential therapeutic target in PE to regulate immune tolerance.

## 1. Introduction

Hypertensive disorders are a common complication in pregnancy, resulting in an increased risk of further complications, as well as long-term consequences for women and their fetuses [1,2,3]. Preeclampsia (PE), one of these hypertensive disorders, represents a severe pregnancy complication affecting 2–5% of all pregnancies [4]. The main diagnostic criteria of PE are based on the following symptoms: a new onset hypertonia (>140/90 mmHg), combined with proteinuria (>300 mg/24 h) or other organ dysfunction in the second half of pregnancy [5]. PE is associated with high morbidity and mortality, causing 70,000 maternal deaths worldwide per year [6].

Due to the lack of a cause-specific therapy for PE [6,7], details about the pathophysiology of this mechanism need to be further elucidated. However, the development of PE cannot be attributed to one particular cause, since numerous factors are involved in the pathophysiology. Currently, it is known that the pathogenesis of PE progresses in two stages, beginning with a defective trophoblast invasion and spiral artery remodeling, as well as immunological alterations in the early materno-fetal environment. Later in pregnancy, the reduced uteroplacental flow promotes the release of proinflammatory chemokines, inducing a systemic inflammation [8]. Overall, the combination of inflammatory processes [4,9,10,11], the loss of the maternal tolerance towards the fetus [3,12,13], and a maternal cardiovascular maladaptation [14,15] are important elements. Further, an impaired trophoblast invasion [16,17]—resulting, among other complications, as a general vascular dysfunction and a deficient remodeling of the spiral arteries [5,18,19]—as well as defective placentation [15,20,21,22] appear to lead to placental insufficiency [23] and the release of vasoactive and pro-inflammatory substances that seem to cause the clinical symptoms.

Different studies underline the importance of the maternal immune system in the pathophysiology of PE, assuming an inadequate immune tolerance towards the semi-allogenic fetus that leads to the abnormal trophoblast invasion [24,25,26]. Furthermore, a shifted cytokine secretion of activated T cells towards the Th1 profile, stimulating a proinflammatory function, has already been shown in PE [27]. Other studies detected an extensive activation of either circulating or decidual neutrophils and monocytes in PE [28,29]. In addition, regulatory T cells (Tregs) have been identified as key players in several processes mediating immune tolerance, as in organ transplantation [30]. They are, moreover, assumed as an important immune cell population for the maintenance of the materno-fetal tolerance via the inhibition of natural killer cells (NK), natural killer T cells (NK-T), and T-lymphocytes [31,32].

The amount of circulating and resident Tregs increases in healthy pregnancies until the end of the 2nd trimester [33]. However, in the case of PE, several studies detected a decrease of Tregs in the maternal peripheral blood during pregnancy [34,35], whereas there is hardly any data about decidual Treg recruitment during PE, and existing data are inconsistent. One study has already shown, that Tregs were less located in the decidua of mice [36]. The mechanism of Treg recruitment in PE has not been fully elucidated. However, in other diseases, such as carcinoma [37,38], auto immune diseases [39], infections [40,41], or implantation [42,43], CCL22, the macrophage derived chemokine, and its receptor, CCR4, are well known for their role in the migration of Tregs [44]. CCL22 is produced by certain types of immune cells, such as macrophages, monocyte-derived dendritic cells [45,46], NK cells, and activated T cells [47].

Gal-2, is a member of the family of carbohydrate-binding proteins that participate in multiple cellular mechanisms, such as cell adhesion and activation, cytokine secretion, immune cell migration, and apoptosis, by binding distinct cell surface or extracellular matrix glycoconjugates [48,49,50,51]. Furthermore, galectins are associated with cell death and growth, as well as with cell differentiation, in addition to their modulatory effect on the immune system by regulating monocytes, macrophages, and CD8^+^-T cells. Gal-2 is able to bind to T cells in a β-galactoside-specific manner to induce apoptosis in activated T cells [52,53,54]. Structurally, Gal-2 is closely related to galectin-1 (Gal-1), although it acts via different cell surface binding strategies [55]. While little is known about the relationship between Gal-2 and Tregs, Gal-1 is considered as a negative regulator of the immune response promoting Treg induction, differentiation, and expansion [56,57]. In PE and other pregnancy diseases, the Gal-2 expression is downregulated in the placental tissue [58], in contrast to the increased Gal-2 level in maternal blood during preeclampsia [59]. Still, there is a lack of research regarding the role of Gal-2 during pregnancy affected by PE.

Therefore, the present study targeted the investigation of Tregs’ recruitment in the decidua of PE-affected pregnancies, as well as their behavioral alterations concerning apoptosis and their chemokine attractant CCL22. Furthermore, the effect of Gal-2 on Tregs in cell culture undergoing apoptosis was observed.

## 2. Results

### 2.1. The Number of Tregs Is Decreased in the Decidua of PE-affected Pregnancies

The number of Tregs was evaluated by immunohistochemical FoxP3 staining. Cells were counted in three randomly selected visual fields of the decidua and the average was calculated.

The number of FoxP3-positive cells was significantly reduced (*p* = 0.046) in PE placentas (1.07 ± 1.203 and a range from 0 to 3.67) compared to control placentas (1.80 ± 1.497 and a range from 0 to 6.33) (Figure 1A–C).

In addition, since the weeks of gestation of the observed PE placentas vary over a wide range, the number of Tregs in early-onset PE (before the 34th week of gestation) and late-onset PE (after the 34th week of gestation) was compared. Although there was a descriptive difference between early-onset PE (1.67 ± 1.553) and late-onset PE (0.84 ± 0.992), this difference was not significant (*p* = 0.218).

Considering only the control placentas, a significantly positive correlation of the number of Tregs and maternal age at birth (*r* = 0.550, *p* = 0.001) was found.

### 2.2. CCL22 Expression Is Increased in PE Compared to Control Placentas

The expression of CCL22, which is known to be involved in Treg recruitment, was evaluated individually for the different tissue parts of the placenta. The staining result of the syncytium, evaluated by the mean IRS (International Remmele Score), showed a significantly increased CCL22 cytoplasmatic expression in the PE placentas (4.00 ± 3.006) compared to the control placentas (2.38 ± 1.688; *p* = 0.013) (Figure 2). Further, an analysis of the mean intensity of the CCL22 staining was performed. There, a significantly increased intensity was measured in PE placentas (0.172 ± 0.0079) compared to the controls (0.161 ± 0.0151; *p* < 0.001).

Furthermore, the mean IRS of the CCL22 staining in the decidual part of the placenta was significantly higher (*p* = 0.006) during PE (2.30 ± 2.693) compared to healthy samples (0.76 ± 1.130) (Figure 3). The mean-intensity-analysis did not show a significant difference between PE and the control group. However, a higher mean intensity of CCL22 was detected in the PE group (0.166 ± 0.0125) compared to the control group (0.161 ± 0.0153). 

Moreover, the expression of CCL22 in the syncytiotrophoblast and in the decidua correlated significantly positive (*r_IRS_* = 0.401, *p_IRS_* = 0.001; *r_intensity_* = 0.442, *p_intensity_* < 0.001), indicating an increased expression in the entire placenta in the case of PE.

In addition, since the weeks of gestation of PE placentas vary over a wide range, the expression of CCL22 in early-onset PE and late-onset PE was compared. Even though there was a descriptive difference between early-onset PE (IRS_syncytium_ = 5.88 ± 4.390, IRS_EVT_ = 2.63 ± 2.504; intensity_syncytium_ = 0.174 ± 0.1016, intensity _decidua_ = 0.170 ± 0.1506) and late-onset PE (IRS_syncytium_ = 3.32 ± 2.056, IRS_EVT_ = 2.18 ± 2.805; intensity_syncytium_ = 0.171 ± 0.0070, intensity _decidua_ = 0.165 ± 0.0116), this difference emerged as insignificant (IRS: *p_syncytium_* = 0.277, *p_EV__T_* = 0.534; intensity: *p_syncytium_* = 0.730, *p_decidua_* = 0.219).

### 2.3. Identification of Decidual Cells Expressing CCL22 as EVT

To investigate the type of decidual cells expressing CCL22, an immunofluorescence staining of CCL22 and CK7 was performed. This staining showed a coexpression of CCL22 and CK7 in all cells stained by anti-CCL22 antibody. Thus, the CCL22-expressing cells in the decidua can be clearly classified as EVT, since there are hardly any other trophoblasts in third trimester placentas and CK7 accounts as a specific trophoblast marker [60]. The result for the double expression in the control and PE placentas was nearly identical (Figure 4).

### 2.4. CCL22 and FoxP3 Are Correlating Positively

The expression of CCL22 and the number of placental Tregs correlated significantly positive in the EVT (*r* = 0.264, *p* = 0.038; Appendix A) but not the syncytiotrophoblast (*r* = 0.239, *p* = 0.061; Appendix A). Individual examination of PE and controls revealed a significantly positive correlation between the number of Tregs and the expression of CCL22 in the syncytium in PE placentas (*r* = 0.576, *p* = 0.001) and a significantly positive correlation of the number of Tregs and the expression of CCL22 in the EVT regarding the control placentas (*r* = 0.465, *p* = 0.006).

### 2.5. Tregs Undergo Apoptosis in PE

To understand the decreased number of FoxP3-positive cells in PE placentas despite increased CLL22 expression, TUNEL staining was performed to identify apoptotic Tregs. A clear difference between the control group and the PE group in the percentage of TUNEL-positive Tregs was found. While only 20–30% apoptotic Tregs appeared in the control group, almost 100% of the detected Tregs were undergoing apoptosis in the PE group (Figure 5).

### 2.6. Correlation of Gal-2 with Tregs

Since galectins are known to be able to induce or inhibit the apoptosis of T cells [55,61,62], data about the expression of galectins—which was detected by our group earlier and published by Hutter et al. [53]—were correlated with the number of Tregs (Appendix A). A significant positive correlation between Gal-2 in the syncytiotrophoblast and the number of decidual Tregs was detected (*r* = 0.390, *p* = 0.049). Considering the control and PE placentas individually, a significantly positive correlation was shown between the number of Tregs and the expression of Gal-2 in PE placentas (Gal-2 in the syncytium: *r* = 0.620, *p* = 0.042; Gal-2 in the decidua: *r* = 0.720, *p* = 0.012).

### 2.7. Gal 2 Protects Tregs from Apoptosis

Since a positive correlation between the number of Tregs and Gal-2 expression, as well as an increased ratio of apoptotic Tregs, was detected in PE placentas, the influence of Gal-2 on the apoptosis of Tregs was investigated. Therefore, the apoptosis in Tregs isolated from blood of healthy patients was induced by FAS ligand (FasL) with and without the addition of Gal-2 and compared with an untreated Treg control group.

Since the results varied highly between the donors, the measured levels of Caspase 3 were set in relation to the Treg + FasL group donor-specifically, further termed as standardized values. The Treg + FasL group was chosen as the reference group, since the highest apoptotic rate was suspected in this group.

Our results showed a significantly reduced amount of active Caspase 3, which is an indicator of active apoptosis, in Tregs incubated with Gal-2 and FasL compared to the group incubated with solely FasL (*p_standardized_* = 0.001, *p_concentration_* = 0.161).

As one donor showed a higher rate of apoptosis in the untreated Tregs than in those treated with FasL, that sample was excluded from the overall statistical analysis, as it can be assumed that the apoptosis induction was defective. Nevertheless, a descriptive analysis showed a reduction in the level of Caspase 3 after addition of Gal-2 in this donor as well (Treg + FasL = 45.87 ± 19.45; Treg + FasL + Gal-2 = 16.99 ± 22.71).

The concentration of active Caspase 3 showed significant differences between the untreated Treg group, the Treg + FasL group, and the Treg + FasL + Gal-2 group (Treg = 81.56 ± 91.979, Treg + FasL = 437.02 ± 43.915, Treg + FasL + Gal-2 = 260.39 ± 147.971; *p* = 0.004). The pairwise comparison of the single groups revealed significant differences between the groups Treg and Treg + FasL (*p* = 0.001) but not between the groups Treg + FasL and Treg + FasL + Gal-2 (*p* = 0.054) (Appendix A).

In contrast, standardized values showed an even higher significance between the untreated control group and the FasL group (*p* ≤ 0.001), as well as significant differences between the groups with and without Gal-2 (*p* = 0.018) (Figure 6). Therefore, the lack of a significant difference in the measured levels of Caspase 3 seems to occur through donor-specific differences.

## 3. Discussion

Various theories for placental dysfunction during the pathophysiology of PE exist: oxidative stress [63,64], generation and transformation of the spiral arteries [65,66], and the imbalance between the maternal adaptive immune system as a proinflammatory response and a lack of immune tolerance towards the semi-allogenic fetus [24,25]. Therefore, the role of the maternal immune response during the development of PE needs to be further investigated to elucidate pathophysiologic mechanisms of PE and to discover potential therapeutic targets. In the present study, we were able to detect a significantly reduced number of FoxP3positive cells, considered to be Tregs, in the decidua of PE-affected pregnancies despite an upregulation of CCL22, a potent Treg chemoattractant. Furthermore, we detected higher rates of apoptotic Tregs in PE placentas. Gal-2, a well-known immunoregulator, which is downregulated in PE placentas, could be identified to protect Tregs from apoptosis in vitro.

Tregs are known to play an essential role in controlling immune regulatory processes. Since there are incoherent findings about Tregs’ recruitment in PE-affected pregnancies and the importance of Tregs during implantation had already been demonstrated in mice [67], the aim of the present investigation was to clarify the aspect of decidual Tregs’ recruitment and to elucidate their role during PE. The present study detected a reduced number of FoxP3-positive cells in the decidua of PE placentas, supporting the findings from previous research that showed reduced levels of circulating and decidual resident Tregs [67,68] in pregnancies suffering from PE. 

However, when interpreting the results, the differing level of Tregs during pregnancy needs to be considered when interpreting our results. Overall, the number of placental Tregs peaks in the second trimester, followed by decreasing values towards the end of pregnancy [33,69,70]. Since the different weeks of gestation in the PE and control groups could be excluded as a cause for the different number of Tregs through regression analysis and pregnancy week matched analysis (which may be seen in the Appendix A), we aimed to analyze the recruitment of Tregs as a possible explanation for the lower number of decidual Tregs in PE, assuming that lower chemoattractant levels might be responsible for the reduced Treg infiltration. Therefore, we chose to analyze CCL22, a well-known chemoattractant for Tregs [71].

Generally, the role of CCL22 during pregnancy is not completely resolved and CCL22 expression in the placental tissue has not been previously investigated during preeclampsia. Still, CCL22 is known to be expressed by dendritic cells and macrophages, both of which account for a large proportion of decidual immune cells [72], as well as in trophoblasts and maternal stromal cells [42]. Macrophages are essential players in remodeling the uterine vasculature, thereby facilitating an adequate placental–fetal blood supply [73,74]. Furthermore, as an immune cell-derived cytokine, CCL22 is involved in M2 polarization of placental macrophages [75]. These immune cells enhance endocytosis and promote tissue repairing mechanisms and cell growth, as well as tissue remodeling. They further promote maternal immune tolerance against the semi-allogenic fetus and preserve fetal growth until delivery [76,77]. Further CCL22 was revealed as a marker for preeclampsia in one study where the serum of pregnant women was analyzed [78]. In accordance with our results, which showed a significantly higher expression of CCL22 in PE placentas, Freier et al. [42] generated the hypothesis of placental CCL22 acting as a negative feedback response to proinflammatory events, since they found no decidual CCL22 expression in healthy first trimester placenta, in contrast to an increased decidual expression in recurrent miscarriages. Since we were able to identify EVT cells expressing CCL22 via immunofluorescence, we could confirm the findings of Freier et al. [42], stating that CCL22 is not only secreted by solid tumor cells, epithelial cells, and immune cells, such as monocytes and macrophages [79,80,81]. Moreover, this could explain the recruitment of Tregs by trophoblast cells, which was already shown by several studies [82,83]. Until now, hCG was assumed to be one of the potential attractants, whereas downregulation of hCG production after siRNA intervention led to reduced Treg recruitment [84]. Nevertheless, the results of the present study hinted at trophoblasts being able to secrete CCL22. However, future research is needed to support this theory, including further confirmation by in vitro and in vivo investigations.

Although we found a diminished number of decidual Tregs but a significantly higher expression of CCL22 in PE placentas, the amount of decidual Tregs correlated significantly positive with the placental CCL22 level. Therefore, the hypothesis of inhibited Treg recruitment by a lack of CCL22 in preeclamptic placenta could be refuted.

Earlier studies had already shown an impaired function of Tregs during PE [85,86]. Furthermore, Zhang et al. [87] found a reduced proliferation of placental Tregs in preeclampsia by analyzing the Ki67 −/+ Tregs in the placenta. Therefore, we hypothesized that the known impaired trophoblast function in PE, as well as the decreased number of Tregs, seemed to be indicative for an increased apoptosis of decidual Tregs in PE placentas. In this context, we investigated the apoptosis of Tregs via TUNEL staining. By analyzing the co-expression of FoxP3, DAPI, and TUNEL staining, we detected an increased number of TUNEL-positive Tregs in PE compared to control placentas. Therefore, enhanced apoptosis of Tregs in preeclamptic placentas might explain the reduced number of decidual Tregs. Increased Treg apoptosis during PE might be a possible explanation for the reduced number of Tregs in PE placentas despite an increased expression of CCL22, one important Treg chemokine. Nevertheless, there are other chemokine ligands, such as CX3CL1 [88,89] CCL3, CCL4, and CCL5 [90,91,92], whose receptors are further expressed by Tregs during pregnancy [93], potentially contributing to chemokine-mediated migration to the decidua.

The imbalance of proinflammatory and anti-inflammatory acting T cells has already been described in pregnancies affected by PE [94]. A reduction or impairment of the immune modulating Tregs induced by apoptosis might be responsible for a lack of immune tolerance against the semi-allogenic fetus. Since the inappropriate and proinflammatory activation of the immune system is thought to play a considerable role in the development of PE [78], the prevention of Treg apoptosis might account for a potential therapeutic target of this pregnancy-associated disease. To identify a potential factor that could help to prevent Treg apoptosis, we chose Gal-2 to perform in vitro experiments. Gal-2, a member of the family of carbohydrate-binding proteins, plays a pivotal role in T cell apoptosis [55,61,62]. Since the expression of Gal-2 is known to be decreased in PE placentas [53], we performed a correlation analysis of Gal-2 and the number of decidual Tregs, which showed a significantly positive correlation. To analyze the possible effect of Gal-2 on Tregs undergoing apoptosis, apoptosis was induced in primary isolated Tregs via FasL. We showed a significantly reduced level of active Caspase 3 in Gal-2 co-cultured cells, indicating a clearly protective effect of Gal-2 on Tregs undergoing apoptosis.

Although the present study revealed promising results, the research conducted also had some limitations that need to be discussed. While in past studies different subtypes of Tregs have been identified in pregnancy and PE, our study only focused on FoxP3-positive cells and did not differentiate between, e.g., iTregs (Helios-) and nTregs (Helios+), while Hsu et al. [50] found no significant difference in decidual FoxP3+ Tregs. Furthermore, to confirm the protection from apoptosis through Gal-2, primary isolated Tregs from healthy control patients were used. Since it was shown that decidual Tregs are phenotypically distinct from peripheral blood Tregs, this difference needs to be considered when interpreting our results [95,96]. Therefore, future research is needed, using Tregs from pregnant women’s blood or directly from the decidual tissue, to confirm that Gal-2 might be a potential therapeutic target for PE and the protecting effect on apoptosis of Treg apoptosis.

## 4. Materials and Methods

### 4.1. Sample Placental Tissue

Tissue samples were collected from the Department of Obstetrics and Gynecology of the University Hospital, LMU Munich, between 2007 and 2019. The collective consisted of 32 PE placentas and 34 control placentas with a mean maternal age of 32.37 ± 5.659 years (range: 17–44 years). The weeks of gestation at birth differed significantly, with a range between 25 and 40 weeks (*p* < 0.001) between PE and controls; therefore, linear regressions were performed. With regard to both the number of Tregs (*p* = 0.339), and the expression of CCL22 (*p_syncytium_* = 0.064; *p_EVT_* = 0.350), no significant impact of the weeks of gestation was detected (Appendix A). Since the weeks of gestation were missing from two PE placentas, whereas Tregs are known to change in number during ongoing pregnancy, we excluded these two PE placentas from the analysis, but separately reported the analysis including the two placentas in the Appendix A. The gender of the newborns was balanced, with 30 female and 31 male newborns (controls: 18 females, 16 males; PE: 12 females, 15 males) (Appendix A).

### 4.2. Immunohistochemistry

The two immunohistochemical stainings were performed according to different protocols. Unless otherwise stated, the work was carried out at room temperature.

#### 4.2.1. FoxP3 Staining

After deparaffinization of the slides in Roticlear (Carlroth, Arlesheim, Switzerland) for 20 min, the endogenous peroxidase was blocked by a 3% H_2_O_2_ methanol mixture. Rehydration by descending alcohol series was followed by demasking via heat pretreatment using a sodium citrate buffer (pH = 6.0). To prevent an unspecific binding and staining, an incubation with Blocking Solution Reagent 1 (ZytoChem Plus HRP Polymer System; Zytomed Systems, Berlin, Germany) for five minutes was performed. Subsequently, the primary antibody Anti-FoxP3 (Abcam, Cambridge, UK; mouse IgG monoclonal, Clone: 236A/E7; dilution 1:300) was applied to incubate for 16 h at 4 °C. After incubation with the primary antibody, a 20 min incubation with Post-Block Reagent 2 (ZytoChem Plus HRP Polymer System; Zytomed Systems, Berlin, Germany) and a 30 min incubation with HRP Polymer Reagent 3 (ZytoChem Plus HRP Polymer System; Zytomed Systems, Berlin, Germany) followed. For visualization, a DAB+ Substrate chromogen system (Dako, Glostrup, Denmark) was applied and the reaction was stopped with distilled water after two minutes. This was followed by a two-minute counterstain with haemalaun and bluing in tap water. The final steps were dehydration through an ascending alcohol series, treatment with Roticlear (Carlroth, Arlesheim, Switzerland), and covering with Eukitt (Merck, Darmstadt, Germany). Between all working steps, the samples were washed with PBS.

#### 4.2.2. CCL22 Staining

The second immunohistochemical staining followed a different protocol, as it was performed in the Institute of Pathology of the University Hospital, LMU Munich. Instead of PBS, TRIS buffer (pH = 7.5) was used for rinsing. The dewaxing was followed by heat pretreatment with Target Retrieval Solution (Agilent Technologies, Santa Clara, CA, USA). After antigen retrieval, the endogenous peroxidase was blocked with 7.5% aqueous hydrogen peroxide. The 20 min incubation with blocking serum (ImmPRESS Reagent Kit Anti-Rabbit IgG; Vector, Burlingame, USA) was followed by a 60 min incubation at RT with the primary antibody (CCL22/MDC, No 500-P107, 1:200; PeproTech, Rock Hill, USA). The sections were then incubated for 30 min with anti-rabbit Ig (ImmPRESS Reagent Kit Anti-Rabbit IgG; Vector Laboratories, Burlingame, CA, USA). Visualization with DAB+ for three minutes and counterstaining with Hematoxylin Gill’s Formula (Vector Laboratories, Burlingame, CA, USA) followed. Aquatex (Merck, Darmstadt, Germany) was used for covering.

### 4.3. Immunofluorescence Staining

To analyze the immunofluorescence staining, the Zeiss Axiophot fluorescence microscope (Zeiss, Oberkochen, Germany) was used in conjunction with the software AxioVision 4.8.1. Each fluorescence staining was performed on a representative portion of 10% of the respective group.

#### 4.3.1. CCL22-CK7 Staining

The immunofluorescence staining procedure was performed in a manner similar to the aforementioned immunohistochemical staining protocol. Deparaffinization in Roticlear for 20 min was followed by rehydration through a descending alcohol series ending in distilled water. Afterwards, antigen retrieval was performed by heat pretreatment in a pressure cooker with Na-citrate-buffer (pH = 6.0) for five minutes. Ultra-Vision block (Thermofisher, Waltham, MA, USA) was applied for 15 min at room temperature. Later, incubation with primary antibodies against CCL22 and CK7 (mouse IgG1 monoclonal, OV-TL 12/30, 1:30; Novocastra Leica Biosystems, Wetzlar, Germany) for 16 h at 4 °C followed. In the next step, the slices were incubated with secondary antibodies for 30 min at RT. The secondary antibodies Cy3-labeled-Goat-anti-rabbit-IgG (1:500; Dianova, Hamburg, Germany) and Alexa-Fluor-488-labeled-Goat-anti-mouse-IgG (1:100; Dianova, Hamburg, Germany) were mixed before application. After drying in the dark, the sections were mounted with mounting medium for fluorescence containing DAPI.

#### 4.3.2. FoxP3-TUNEL Staining

To perform the double staining of Anti-FoxP3- (mouse IgG1, monoclonal 236A/E7, 1:50; Thermofisher, Waltham, MA, USA) and TUNEL staining, another protocol was necessary. The procedure resembles immunohistochemical FoxP3 staining, although the endogenous peroxidase was not blocked. After heat pretreatment, unspecific binding sites and staining were blocked by incubation with Ultra-Vision-Protein-Block (Thermofisher, Waltham, MA, USA) for 15 min. The sections were then incubated with the primary antibody FoxP3 for 16 h at 4 °C. After washing with PBS, the secondary antibody goat-anti-mouse-IgG Cy3-labeled (Jackson Immunoresearch Laboratories, West Grove, PA, USA) was applied for 30 min at room temperature. In the next step, the TUNEL staining was performed. TUNEL enzyme (Roche, Basel, Switzerland) and TUNEL label (Roche, Basel, Switzerland) were mixed in a ratio of 1:10 and 50 µL were applied on each slide. Covered with a cover glass, the sections were incubated for one hour at 37 °C. After incubation and washing in PBS, the sections were air-dried and covered with mounting medium for fluorescence with DAPI (Vector, Burlingame, CA, USA).

### 4.4. Evaluation of Stainings

Different methods were used to evaluate the staining, depending on the used antibody. The CCL22 staining was primarily evaluated by two independent evaluators, using the semi-quantitative International Remmle Score (IRS). The IRS was calculated by multiplying the percentage of positively stained cells (0 = no staining, 1 ≤ 10%, 2 = 11–50%, 3 = 51–80%, 4 > 80% of cells stained) in the examined tissue type, and the staining intensity (0 = no staining, 1 = weak staining, 2 = moderate staining, 3 = strong staining). The IRS was determined separately for the syncytium and the decidua, with the entire slide being evaluated. In addition, a software-related evaluation was performed using the open-source software QuPath (version 0.3.0; Github, San Francisco, CA, USA). For this purpose, three images were taken of both the syncytium and the decidua of the slide under investigation at a 6.3× magnification (Flexcam C1, Leica microsystems, Wetzlar, Germany). Subsequently, the sole DAB staining was isolated in each image and the mean intensity per pixel was analyzed. For the overall analysis, the mean value for each of the three images per tissue type was calculated.

For the analysis of Tregs, the FoxP3-positive cells were counted in three randomly selected visual fields of the decidua at 25× magnification in the immunohistochemical staining and at 20× magnification in the immunofluorescence staining and the average was calculated.

### 4.5. Cell Culture of Tregs and Gal-2

Tregs were isolated from PBMC (peripheral blood mononuclear cells) of human donor blood with the MACS CD4+ CD25+ CD127dim/− human regulatory T cell isolation kit II human (Nr. 130-094-775, Miltenyi Biotec, Bergisch Gladbach, Deutschland). The purity of Tregs was verified by flow cytometry analysis using the BD LSRFortessa Flow Cytometer (Becton Dickinson, Franklin Lakes, NJ, USA) (Appendix A). The antibodies that were used for the FACS verification are listed in Appendix A. The cell culture was performed as a biological triplicate, meaning that the Tregs were isolated from buffy coats of three different healthy donors. Basically, 200,000 freshly isolated Tregs were seeded in one ml RPMI-1640-medium per well of a 24-well plate. The effect of Gal-2 on the isolated and FasL-pretreated Tregs was analyzed. Two groups of Tregs were seeded for this purpose, in addition to the control group with untreated Tregs. In the first group, apoptosis was induced by using one µg/mL FasL (Treg + FasL); in addition, one µg/mL Gal-2 was added to one µg/mL FasL (Treg + FasL + Gal-2) for the second group. After five hours of incubation, the cells were extracted for the subsequent caspase-3 ELISA.

### 4.6. Caspase 3 ELISA

To analyze the amount of apoptotic Tregs after cell culture with Gal-2, the amount of active Caspase 3 was measured after cell extraction with the human active Caspase-3 immunoassay Quantikine ELISA (R&D Systems, Minneapolis, MN, USA). For a more accurate result, each sample was analyzed in technical triplicates.

### 4.7. Statistical Analysis

Statistical analysis was performed using the PC software SPSS (version 24; IBM, Armonk, NY, USA). Non-parametric tests, such as the Mann-Whitney U-test and the Spearman-Rho correlation test, were used, as the values could not be assumed to have a normal distribution. The results are given as mean value ± standard deviation. The correlation coefficient *r* indicates the strength of the correlation (*r* < 0.3 weak relation, *r* > 0.3 medium relation, *r* > 0.5 strong relation) [97]. In order to analyze a possible effect of the weeks of gestation, a linear regression was performed. The significance level for all tests was assumed at *p* < 0.05. In addition, an analysis of matched data regarding the weeks of gestation was conducted, using the Wilcoxon rank test, as shown in the Appendix A.

## 5. Conclusions

In summary, our results show that Tregs undergo apoptosis during PE, which may be prevented by Gal-2. Furthermore, we detected an increased expression of CCL22 in PE placentas, while Treg infiltration was reduced, indicating a positive feedback loop. Whether Gal-2 might be a potential therapeutic target to avoid Treg apoptosis, and therefore prevent an immunomodulatory imbalance during PE, needs to be further investigated in additional research.

## Figures and Tables

**Figure 1 ijms-23-01880-f001:**
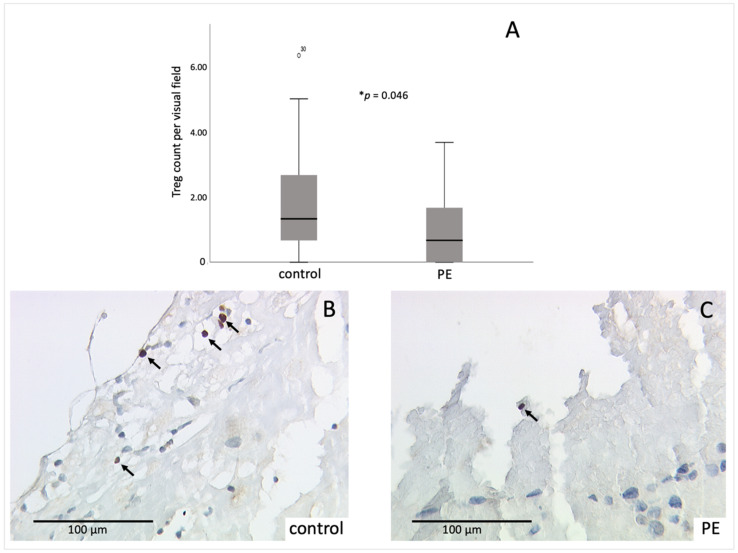
Immunohistochemical staining results of FoxP3: (**A**) boxplot of the average number of FoxP3-positive Tregs per visual field in control and PE placentas (control *n* = 34, PE *n* = 30), mean ± SD; *p*-values were calculated with Mann-Whitney-U-Test, * *p* = 0.046; (**B**) representative picture of control placenta; (**C**) representative picture of PE placenta. Detected Tregs are marked with arrows. The circle in **A** symbolizes an outlier value with its respective number for identification. A respective negative and positive control is shown in the Appendix A.

**Figure 2 ijms-23-01880-f002:**
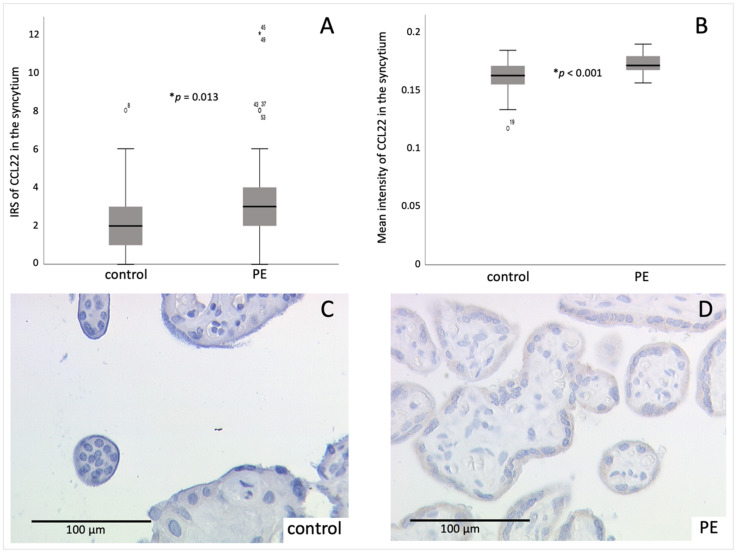
Immunohistochemichal staining results of CCL22 in the syncytium: (**A**) boxplot of the mean IRS of syncytial staining of CCL22 in control and PE placentas (control *n* = 34, PE *n*= 30), mean ± SD; *p*-values were calculated with Mann-Whitney-U-Test, * *p* = 0.013; (**B**) boxplot of the mean intensity of syncytial staining of CCL22 in control and PE placentas (control *n* = 34, PE *n* = 30), mean ± SD; *p*-values were calculated with Mann-Whitney-U-Test, * *p* < 0.001; (**C**); representative picture of control placenta (IRS = 3); (**D**) representative picture of PE placenta (IRS = 12). The circles in **A** and **B** symbolize outlier values with their respective number for identification. The star in **A** symbolizes an extreme outlier value with its respective number for identification. A respective negative and positive control is shown in the Appendix A.

**Figure 3 ijms-23-01880-f003:**
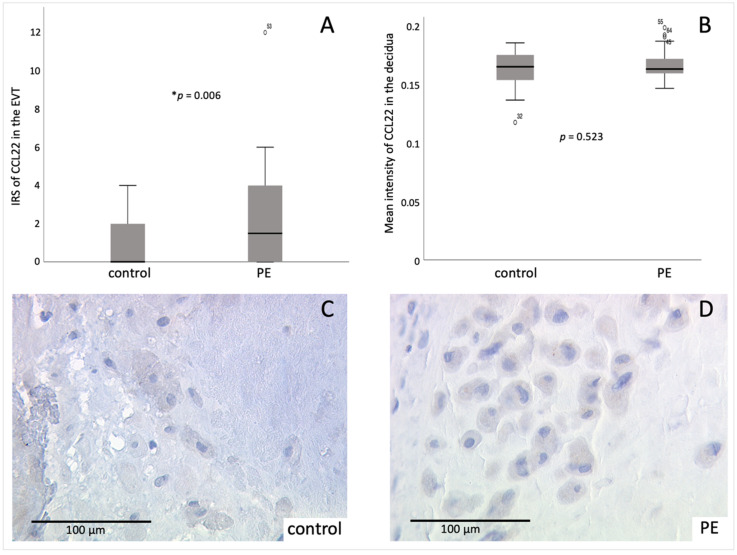
Immunohistochemical staining results of CCL22 in the EVT (extravillous trophoblast): (**A**) boxplot of the mean IRS of decidual staining of CCL22 in control and PE placentas (control *n* = 34, PE *n* = 30), mean ± SD; p-values were calculated with Mann-Whitney-U-Test; * *p* = 0.006 (**B**) boxplot of the mean intensity of decidual staining of CCL22 in control and PE placentas (control *n* = 34, PE *n* = 30), mean ± SD; *p*-values were calculated with Mann-Whitney-U-Test; (**C**) representative picture of control placenta (IRS = 3), (**D**) representative picture of PE placenta (IRS = 6). The circles in (**A**) and (**B**) symbolize outlier values. Respective negative and positive control is shown in the Appendix A.

**Figure 4 ijms-23-01880-f004:**
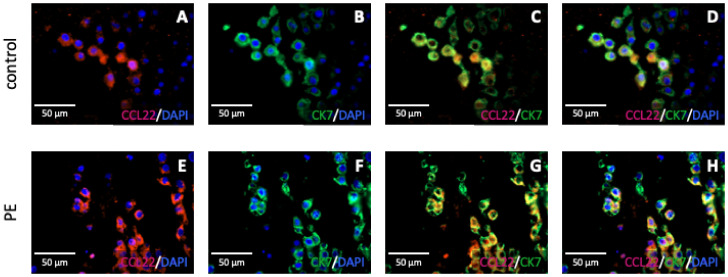
Immunofluorescence staining of CCL22 (red) and CK7 (green) in the decidua of PE and control placentas. Nuclear staining with DAPI is shown in blue in each case. Representative pictures of control placenta (**A**–**D**) and PE placenta (**E**–**H**), single staining of CCL22 (**A**,**E**) and CK7 (**B**,**F**), double staining of CCL22 and CK7 (**C**,**G**), and merge including nuclear staining (**D**,**H**) are shown. Respective negative control picture is shown in the Appendix A.

**Figure 5 ijms-23-01880-f005:**
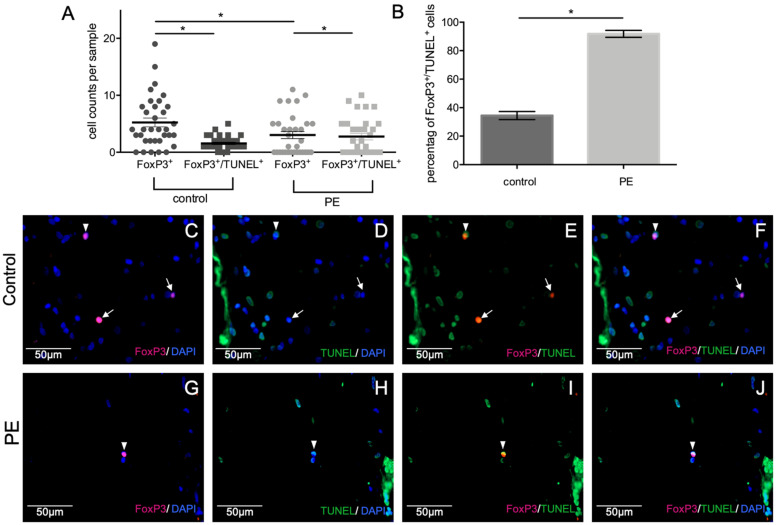
Immunofluorescence staining of apoptotic Tregs. Graphical representation of all Tregs (FoxP3^+^) and all apoptotic Tregs (FoxP3^+^/TUNEL^+^) counted per sample (**A**), mean ± SD; * *p*-values were calculated with Mann-Whitney-U-Test; mean percentage of apoptotic Tregs counted in controls and PE (**B**), mean ± SD; * *p*-values were calculated with Mann-Whitney-U-Test; Representative pictures of control placenta (**C**–**F**) and PE placenta (**G**–**J**), single staining of FoxP3-positive Tregs (red) (**C**,**G**), single TUNEL staining (green) (**D**,**H**), double staining of apopototic Tregs (**E**,**I**) (yellow), merge (**F**,**J**), and nuclear staining with DAPI (**C**,**D**,**F**,**G**,**H**,**J**). Non-apoptotic Tregs are marked with arrows; apoptotic Tregs are marked with arrowheads. A respective negative control is shown in the Appendix A.

**Figure 6 ijms-23-01880-f006:**
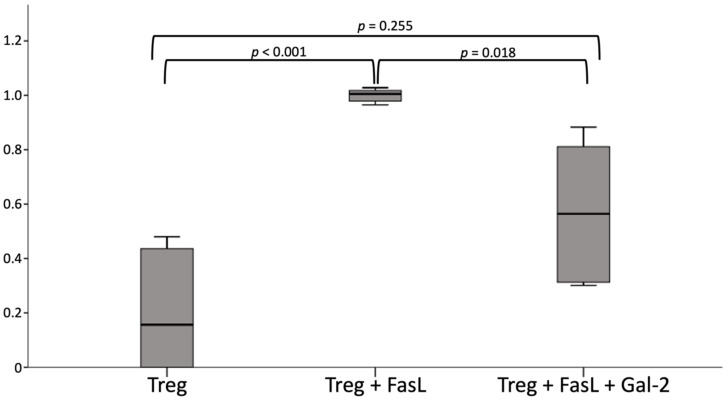
Concentration of standardized active Caspase 3 after exclusion of the second donor, in the groups untreated Treg (Treg, 0.20 ± 0.231), Treg with induction of apoptosis through FasL (Treg + FasL, 1.00 ± 0.024), and the group with the Gal-2 treatment (Treg + FasL + Gal-2, 0.57 ± 0.283). The one-way-ANOVA-Kruskal-Wallis-Test showed significant differences among the three groups (*p* = 0.002). Further analysis revealed this significant difference between the group Treg and Treg + FasL (*p* < 0.001), as well as between Treg + FasL and the group Treg+FasL+Gal2 (*p* = 0.018).

## Data Availability

The data presented in this study are available on request from the corresponding author. The data are not publicly available to ensure privacy of the participants.

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
