# Peer review of "Regulatory T Cell Apoptosis during Preeclampsia May Be Prevented by Gal-2"

_ijms, 2022, doi:10.3390/ijms23031880_

Round 1

Reviewer 1 Report

Minor comments

1- What is the  source of Gal-2?? Bacterial recombinant protein??  If yes,   it should be  endotoxin free 

2- I suggest not to include the  session 4.7 and 4.7.1 in the methods because the authors did not present any data on CLL22 mRNA expression  neither in the manuscript nor in the supplements. 

Author Response

Reviewer 1:

Thank you very much for appreciating our study results and manuscript. We tried to follow your comments and your suggestions.

Minor comments

1- What is the  source of Gal-2?? Bacterial recombinant protein??  If yes, it should be  endotoxin free 

The source of Gal-2 is E. coli. The used Gal-2 is according to its data sheet “a recombinant protein with a N-Terminal His-tag and corresponding to the amino acids 1-132 of Human Galectin-2”. We added the data sheet at the end of this letter. It should be endotoxin free. For your reference the link tot the data sheet of the company: https://www.novusbio.com/products/galectin-2-recombinant-protein_nbp1-48326

2- I suggest not to include the  session 4.7 and 4.7.1 in the methods because the authors did not present any data on CLL22 mRNA expression  neither in the manuscript nor in the supplements. 

Thank you for your comment. You are of course correct, we have to apologize for this error, we have deleted the section from the manuscript.

Reviewer 2 Report

In this manuscript Meister and colleagues analysed the expression of regulatory T-cells (Tregs), key players in immune tolerance, in normal and preeclamptic pregnancies. They identified Tregs in the decidua by the expression of FoxP3 finding a reduced number of FoxP3 positive cells in the decidua of PE placentas. Furthermore, they found increased expression of CCL22 at the maternal-fetal interface of pregnancies complicated by PE. the authors concluded that Tregs undergo apoptosis at the materno-fetal interface during PE.

The manuscript needs substantial clarification. In particular: 

  • Line 50-53:  The authors should stress the complexity of PE pathology since it has been also reported that PE is also characterised by trophoblast immaturity (PMID: 32529396) and vascular dysfunction (PMID: 34831277).  
  • Line 96: "decreased in in the decidua" there is a typing error
  • Line 103-107:FoxP3 expression must be compared only with age matched controls. There must not be differences in gestatinal age between PE (late or early) and control pregnancies analysed otherwise the differences that can be seen may not be due to the pathology but to the different gestational ages
  • Figure 1 and 2: negative controls for FoxP3 and CCL22 must be shown 
  • Figure 4 and 5: negative controls for FoxP3 and CCL22 must be shown 
  • A table of clinical characteristics of patients must be reported. The table S1 provided by the authors is incomplete. Maternal ages, birth weight and gestational ages at delivery must be reported. The mean gestational age of PE and Control group cannot be different otherwise the differences found in FoxP3 and CCL22 expression may be not due to the pathology but to the different gestational age between the groups. Samples with unknown gestational age cannot be included in the study. 
  • An accurate revision of typing and punctuation error must be done

Author Response

Reviewer 2:

In this manuscript Meister and colleagues analysed the expression of regulatory T-cells (Tregs), key players in immune tolerance, in normal and preeclamptic pregnancies. They identified Tregs in the decidua by the expression of FoxP3 finding a reduced number of FoxP3 positive cells in the decidua of PE placentas. Furthermore, they found increased expression of CCL22 at the maternal-fetal interface of pregnancies complicated by PE. the authors concluded that Tregs undergo apoptosis at the materno-fetal interface during PE.

Thank you very much, we appreciate reviewer´s suggestions and tried to refer to your comments and concerns to improve our manuscript.

The manuscript needs substantial clarification. In particular: 

Line 50-53:  The authors should stress the complexity of PE pathology since it has been also reported that PE is also characterised by trophoblast immaturity (PMID: 32529396) and vascular dysfunction (PMID: 34831277).  

Thank you for your comment. We added some information of further pathophysiological causes of PE to the introduction and cited both publications. Please refer to line 71-78 in the manuscript.

Line 96: "decreased in in the decidua" there is a typing error

Thank you for your comment. You are of course correct, we apologize for this and have corrected the typing error.

Line 103-107:FoxP3 expression must be compared only with age matched controls. There must not be differences in gestatinal age between PE (late or early) and control pregnancies analysed otherwise the differences that can be seen may not be due to the pathology but to the different gestational ages

Thank you for your comment. We agree with you that, the number of Treg changes during pregnancy, especially after the 2nd trimester, and therefore the validity of our data might seem to be reduced. However, several points remain to be considered. First, we performed a regression analysis which shows, that at least in our collective there is no significant correlation with the week of pregnancy. On the other hand, it has to be considered that according to the literature the number of Treg decreases with increasing gestational week, which means that in the control group, which has an average higher gestational week, a lower number of Treg would be expected than in the group of preeclampsia placentas with a lower gestational week. However, because we found significantly more Treg in the controls, we would expect to find an even more pronounced difference in a well-matched population.

Nonetheless, we excluded the two PE placentas with missing gestational weeks from the analysis, as you wished and added the statistical analysis of the pregnancy week matched data in the supplementary data (Supp. Table. S3).

Figure 1 and 2: negative controls for FoxP3 and CCL22 must be shown 

We have added the corresponding negative controls in the supplementary data and refer to them in the picture caption. (Supp. Fig. S1 and S2) If desired, we can alternatively add the negative controls in the image itself.

Figure 4 and 5: negative controls for FoxP3 and CCL22 must be shown 

We have added the corresponding negative controls in the supplementary data and refer to them in the picture caption (Supp. Fig. S3). If desired, we can alternatively add the negative controls in the image itself.

A table of clinical characteristics of patients must be reported. The table S1 provided by the authors is incomplete. Maternal ages, birth weight and gestational ages at delivery must be reported. The mean gestational age of PE and Control group cannot be different otherwise the differences found in FoxP3 and CCL22 expression may be not due to the pathology but to the different gestational age between the groups. Samples with unknown gestational age cannot be included in the study. 

Thank you for your comment. We are pleased to provide you with the table completed by the corresponding clinical data. Unfortunately, we were not able to find out all corresponding data for every placenta included in our analysis, due to partial lack of proper documentation. We excluded two PE placentas from our analysis where the week of pregnancy could not be found.

An accurate revision of typing and punctuation error must be done

Thank you for your comment. We have conducted an exact revision of typing and punctuation errors and hope we have corrected all errors.

Round 2

Reviewer 2 Report

The revised manuscript is largely improved and can be accepted in the present form 

This manuscript is a resubmission of an earlier submission. The following is a list of the peer review reports and author responses from that submission.

Round 1

Reviewer 1 Report

The authors provided an interesting paper focused on studying  the recruitment  of Treg cells to the decidua of PE affected pregnancies,  their apoptotic status and tested  ( in vitro ) the contribution of  Galectin 2 as a potential anti-apoptotic protein during preeclampsia  

# Figures:  The cohort includes 32 PE and 34 Control placentas. It is not clear how many placentas/ deciduas were tested and included in statistic analyses.  . It is suggested that  the number of samples   ( N) should be included in  the figures that shows statistic data ( figure 1,2,3 )

# 2.7 Correlation of Gal-2 with Tregs

The data presented under this subsession is not supported by figure, table, how it was tested etc. Data should be shown 

# The  lack of information about the source of the Gal-2 is a real concern. Is it recombinant?? expression sytem?? Purity??

It is well known that nanogram quantities of endotoxin can affect the biochemical events of certain cell and can interfere with in vitro experiments. I recommend that  authors exclude the presence of endotoxin in the Gal-2 

# Line 197: The author mentioned that"  the lack of significant differences in the .........seems to occur through donor specific difference ". The results of Capase activity is relying on 3 blood samples. I think that there is a place to increase the no. of samples in order to increase the statistic power . 

# It is not clear if the authors included a control  galectin ( or other protein) as a negative  control  on apoptosis induced by FasL   ( Figure 6 and  2.8).

# The cohort includes early and late PE with GA ranging from 25-40 weeks. Is there any differences in Treg level , CCL2 and and appototic Treg cells in eraly vs late PE??. The author should  consider and discuss this point .

Author Response

Reviewer 1:

The authors provided an interesting paper focused on studying  the recruitment  of Treg cells to the decidua of PE affected pregnancies,  their apoptotic status and tested  ( in vitro ) the contribution of  Galectin 2 as a potential anti-apoptotic protein during preeclampsia  

Thank you very much for appreciating our study results and manuscript. We tried to follow your comments and your suggestions.

# Figures:  The cohort includes 32 PE and 34 Control placentas. It is not clear how many placentas/ deciduas were tested and included in statistic analyses.  It is suggested that  the number of samples   ( N) should be included in  the figures that shows statistic data ( figure 1,2,3 )

We apologize that you considered the inclusion of the cohort in the statistical analysis as unclear. For the immunohistochemical analyses, the entire cohort of each of the 32 PE and 34 control placentas were included. We have added the information to the figure legends of Figures 1-3 according to your request.

# 2.7 Correlation of Gal-2 with Tregs: The data presented under this subsession is not supported by figure, table, how it was tested etc. Data should be shown 

We have added a graphic in the supplementary material. The statistical test used to calculate the correlation can be found in 4.7 statistical analysis and was the Spearman-Rho correlation test for non-parametric tests.

# The  lack of information about the source of the Gal-2 is a real concern. Is it recombinant?? expression sytem?? Purity??

It is well known that nanogram quantities of endotoxin can affect the biochemical events of certain cell and can interfere with in vitro experiments. I recommend that  authors exclude the presence of endotoxin in the Gal-2 

Thank you very much for this important comment. We have added the information regarding the company of recombinant Gal-2 in the manuscript. We are also pleased to provide you with the company's datasheet for the purity of recombinant Gal-2. https://www.novusbio.com/products/galectin-2-recombinant-protein_nbp1-48326

We did not perform any endotoxin measurement, but we would not expect a presence of endotoxins in the selected Gal-2, since it was tested by the company and endotoxins would furthermore influence the survival of the Tregs negatively. I hope you find our argumentation sufficient.

In addition to the groups mentioned in the manuscript, we also performed a measurement of caspase-3 after solitary incubation with Gal-2. However, since it was not possible for us to investigate this group for each donor. Due to the low gain of Tregs, per donor it was not possible to perform this for every donor, therefore it was not mentioned in our manuscript so far.

A representative sample showed no difference between the untreated control group and the Tregs incubated with Gal-2 alone.

# Line 197: The author mentioned that"  the lack of significant differences in the .........seems to occur through donor specific difference ". The results of Capase activity is relying on 3 blood samples. I think that there is a place to increase the no. of samples in order to increase the statistic power . 

Since the isolation process of Tregs is quite complex and cell yield is not very high in every case, several studies where Tregs were isolated for cell culture experiments, performed merely technical triplicates, no biological triplicates, or samples were pooled. Although there is no specific source of literature for it, analysis of a wide variety of papers shows that working as a biological triplicate in cell culture is a long-established scientific standard. Furthermore, you are of course correct that a larger sample would result in a greater statistical power. A higher statistical power would only lead to a smaller type II error (beta-error). Your concern, however, refers to the type I error. However, the type I error is determined by setting the significance level. By adding more donors, there will still be big differences between the donors, which would be not easy to be released. Therefore, we thought a standardization would be necessary in every case. We hope we could take your doubts and that you find our argumentation traceable.

# It is not clear if the authors included a control  galectin ( or other protein) as a negative  control  on apoptosis induced by FasL   ( Figure 6 and  2.8).

We are sorry that we have not expressed ourselves clearly enough. We studied three groups: untreated Tregs to serve as controls, Tregs with FasL to quantify the clear induction of apoptosis as well as its level, and the group of Tregs with FasL and Gal-2. We did not examine other galectins in conjunction with FasL since results of older imunnohistochemical evaluations of our group analyzing several other galectins in the placenta, did not show a correlation with the number of Tregs, therefore we felt that this would not have added value to our question. Please let us know whether you find our argumentation acceptable.

# The cohort includes early and late PE with GA ranging from 25-40 weeks. Is there any differences in Treg level , CCL2 and and appototic Treg cells in eraly vs late PE??. The author should  consider and discuss this point .

You are absolutely right, that a comparison of EOP and LOP can offer interesting insights. Since there was no significant difference between EOP and LOP, we had previously excluded this point. We have mentioned that we did not find any difference in our results in our manuscript and added some thoughts about the pregnancy week in our discussion.

Reviewer 2 Report

In this study the investigators analysed the Treg compartment in the placenta during preclampsia (PE). Although the question is relevant the data are not convincing and the methodology very limited and mainly based on IHC and IF analysis in situ with semis quantitative analyses.

Major concerns:

  1. Other techniques should be used to provide more robust quantified data:
    1. RNAseq data on the series of sample analysed would strengthen the differences between PE and controls and the various correlations (Treg and CCL22, and GAL2 for example)
    2. Quantification of CCL22 need to be quantify through alternative methods, really quantitative at least through image analysis software.
  2. Major problem: the placenta from PE are certainly from earlier gestation than the control, and may reflect the differences between in Treg frequency between PE and controls. This is not specified in the m&m.
  3. Fig.5 :Apoptosis of Treg is not convincing at all: only one cell shown in panel A-H!!, no quantification, this is just not scientifically acceptable
  4. Correlation with Gal2 and Treg should be shown. How Gal2 is detected and quantified.
  5. The plot for CCL22 and Foxp3 correlation should be shown, even if statistically significant (p=0.04) the correlation is very weak (r=0.2) and not convincing.

Minor concerns:

  1. Fig.1 A wider field should be shown in panel B/C
  2. Fig.2/3: what about the other CCR4 ligand CCL17?,
  3. Expression of CKR on placenta Treg, and other Treg chemokines in the placenta?

Author Response

Reviewer 2:

In this study the investigators analysed the Treg compartment in the placenta during preclampsia (PE). Although the question is relevant the data are not convincing and the methodology very limited and mainly based on IHC and IF analysis in situ with semis quantitative analyses.

Thank you very much, we appreciate reviewer´s suggestions and tried to refer to your comments and concerns to improve our manuscript.

Major concerns:

  1. Other techniques should be used to provide more robust quantified data:
    • RNAseq data on the series of sample analysed would strengthen the differences between PE and controls and the various correlations (Treg and CCL22, and GAL2 for example)

Thank you for this comment. This is one of our aims we had but so far samples which were collected were not appropriate for RNAseq analysis therefore we optimized the collection for future experiments. We furthermore concentrated now on macrophages which should be the main carrier of CCL22 in the placenta. We were able to isolate macrophages out of fresh placental tissue and managed to get good enough RNA quality out of some cells and plan to do a RNAseq analysis in the near future.

1.2 Quantification of CCL22 need to be quantify through alternative methods, really quantitative at least through image analysis software.

We have tried several methods for the evaluation of CCL22 in the placenta. Several different antibodies were tried to establish a western blot for CCL22 out of frozen placenta tissue, without success, further a PCR out of frozen placenta tissue was performed to get representative quantitative results but the very low amount of CCL22 in the placental tissue did not allow to get clear results. Therefore, we decided to use the internationally recognized IRS for the pathologic analysis of the expression of proteins in the tissue stained by IHC for our analysis. We also tried to do analysis of CCL22 by quantitative software, but there it was only possible to analyze the intensity of selective cells and not of the whole tissue to find out a representative expression level of CCL22. IRS was performed double blinded by two independent evaluators to get objective results.

  1. Major problem: the placenta from PE are certainly from earlier gestation than the control, and may reflect the differences between in Treg frequency between PE and controls. This is not specified in the m&m.

Thank you very much for this comment, it is definitely necessary to consider this when interpreting our results.

However, it is shown in the literature that the number of Treg peaks in the second trimester and decreases in the further pregnancy until birth (Figueiredo & Schumacher, 2016; Hosseini, Dolati, Hashemi, Abdollahpour-Alitappeh, & Yousefi, 2018; Robertson et al., 2019; Teles, Zenclussen, & Schumacher, 2013). Therefore, if one assumes the significant difference of Treg cell number in the two groups is because of the differing pregnancy weeks contradictory results would be expected. However, this is definitely an important aspect to discuss in our manuscript, therefore we added corresponding content in our discussion. Further we were able to show in the regression analysis which we performed, that the week of gestation had no significant influence on the results of our investigations, neither on the number of Tregs, nor on the expression of CCL22. Please refer to corresponding graphs in updated supplement.

  1. Fig.5 : Apoptosis of Treg is not convincing at all: only one cell shown in panel A-H!!, no quantification, this is just not scientifically acceptable

Figure 5 was only intended as a representative image to show a field of view from the placentas examined. Due to the greatly reduced number of Tregs in the preeclampsia placentas compared to the controls, it was not possible to display more than one Treg on one image which shows still recognizable Tregs. The number of decidual Tregs in the third trimenon is as low as shown in Fig. 1. Analysis was performed with a 40x objective in one visual field. Showing more Tregs in one visual field would only be possible if a disproportionately small magnification was selected, in which, however, double staining would no longer be easily recognizable. When counting all Tregs in the decidua which was shown in each sample there were not more than 20 cells in the control placentas, while in the PE placentas only 11 Treg could be found. In addition this is also represented in the mean count of Treg per visual field shown in the immunohistochemistry (2.1). However, in the images E-H more than one single cell can be detected, but since these cells are not Treg, they are only stained with DAPI in blue (this is a cell nucleus staining) or partly additionally by TUNEL in green. Furthermore, we have added further pictures for you, in a wider field of magnification. If you wish, we can add these pictures in our supplement e.g..

Staining colors: blue = DAPI for nucleus staining, red = FoxP3, green = TUNEL

Example of a wider field of magnification in a control placenta. As you can see the smaller the magnification, the harder it gets to distinct a Treg from a FoxP3 stained point that is not considered to be a cell (no DAPI-Staining).

Staining colors: blue = DAPI for nucleus staining, red = FoxP3, green = TUNEL

Example of a wider field of magnification in a PE placenta. Caused by the small amount of Tregs found in the PE placentas in a big amount of fields of views, no Tregs could be detected at all. Therefore, the single Treg represented in the pictures E-H in Figure 5 are not caused by a non-scientific way of work but through the low density of Tregs in the PE placentas.

Staining colors: blue = DAPI for nucleus staining, red = FoxP3, green = TUNEL

Example of a PE placenta with an even higher magnification than shown in the manuscript. Here you can see one apoptotic Treg as well as another apoptotic cell that isn’t FoxP3 positive. We think this is a good picture to distinct between the different stained cells, but it does not provide a good overview over the placental tissue.

  1. Correlation with Gal2 and Treg should be shown. How Gal2 is detected and quantified.

Thank you for this note. We are pleased to explain this better in our manuscript and to you. The Gal-2 expression data that we used for the calculation of correlation with Treg were studied earlier in our laboratory and published by Hutter et al. Gal-2 was also immunohistochemically stained and analyzed by IRS in this study.

  1. The plot for CCL22 and Foxp3 correlation should be shown, even if statistically significant (p=0.04) the correlation is very weak (r=0.2) and not convincing.

 We agree with you that the data needs to be shown, thank you for this, we have added the plot for the correlation. Since according to Cohen (1977) a correlation between r = 0.3 and r = 0.5 is considered to be moderate, our correlation of r = 0.25 may be weak, but in a statistical point of view every significant correlation is a relevant correlation even if it is weak. Moreover, the correlation between Treg and CCL22 is already a well-known and variously proven correlation, which is why we assume this correlation is convincing, especially in combination with our other results. We really hope you find our argumentation appropriate and traceable.

Minor concerns:

  1. Fig.1 A wider field should be shown in panel B/C

We apologize that we can’t follow this wish of the reviewer. We would like to maintain the representative pictures of our analysis which was performed with a 40x magnification and cells were counted per visual field and this is shown in our representative pictures. Further, please refer to our explanation concerning the “density” of Tregs in the placental decidua in term placenta. If demonstrating the cell number in a lower magnification, cells would not be identifiable as positive Tregs and analysis would be limited.

  1. Fig.2/3: what about the other CCR4 ligand CCL17?,

You are absolutely right that there are a variety of chemokines and receptors that influence Tregs. We also mention in our discussion that the influence of other chemokines needs to be evaluated in future studies. In our study, we chose to evaluate CCL22 initially because it is one of the best studied Treg-attracting chemokines.

  1. Expression of CKR on placenta Treg, and other Treg chemokines in the placenta?

We agree with you that it would be very interesting to find out more about the different chemokines of Tregs in the placenta. This will be part of our further studies, since we concentrated more on CCL22 as major Treg chemoattractant, the apoptosis of Tregs and the influence of Gal-2 in our study. But we are about to perform an analysis of CCR4 in our collective, results are awaited in the next months.

Cohen, J. (1977). Statistical power analysis for the behavioral sciences. New York: Academic Press.

Figueiredo, A. S., & Schumacher, A. (2016). The T helper type 17/regulatory T cell paradigm in pregnancy. Immunology, 148(1), 13-21. doi:10.1111/imm.12595

Hosseini, A., Dolati, S., Hashemi, V., Abdollahpour-Alitappeh, M., & Yousefi, M. (2018). Regulatory T and T helper 17 cells: Their roles in preeclampsia. J Cell Physiol, 233(9), 6561-6573. doi:10.1002/jcp.26604

Robertson, S. A., Green, E. S., Care, A. S., Moldenhauer, L. M., Prins, J. R., Hull, M. L., . . . Dekker, G. (2019). Therapeutic Potential of Regulatory T Cells in Preeclampsia-Opportunities and Challenges. Front Immunol, 10, 478. doi:10.3389/fimmu.2019.00478

Teles, A., Zenclussen, A. C., & Schumacher, A. (2013). Regulatory T cells are baby's best friends. Am J Reprod Immunol, 69(4), 331-339. doi:10.1111/aji.12067

Round 2

Reviewer 2 Report

No more comments to the authors, I maintain may multiple concerns.

Author Response

Reviewer 2:

In this study the investigators analysed the Treg compartment in the placenta during preclampsia (PE). Although the question is relevant the data are not convincing and the methodology very limited and mainly based on IHC and IF analysis in situ with semis quantitative analyses.

Thank you very much, we appreciate reviewer´s suggestions and tried to refer to your comments and concerns to improve our manuscript.

Major concerns:

  1. Other techniques should be used to provide more robust quantified data:
    • RNAseq data on the series of sample analysed would strengthen the differences between PE and controls and the various correlations (Treg and CCL22, and GAL2 for example)

Thank you for this comment. This is one of our aims we had but so far samples which were collected were not appropriate for RNAseq analysis therefore we optimized the collection for future experiments. We furthermore concentrated now on macrophages which should be the main carrier of CCL22 in the placenta. We were able to isolate macrophages out of fresh placental tissue and managed to get good enough RNA quality out of some cells and plan to do a RNAseq analysis in the near future.

1.2 Quantification of CCL22 need to be quantify through alternative methods, really quantitative at least through image analysis software.

We now performed an intensity analysis with an image analysis software to evaluate the main intensity of the DAB-Signal in our slides and performed this for every sample. We added these results in our manuscript. We have tried several other methods for the evaluation of CCL22 in the placenta. Several different antibodies were tried to establish a western blot for CCL22 out of frozen placenta tissue, without success, further a PCR out of frozen placenta tissue was performed to get representative quantitative results but the very low amount of CCL22 in the placental tissue did not allow to get clear results, but could confirm the same tendency of CCL22 regulation in our results. Nevertheless, we provided the data in the supplement. We chose the internationally recognized IRS as semi quantitative method for the pathologic. IRS was performed double blinded by two independent evaluators to get objective results.

  1. Major problem: the placenta from PE are certainly from earlier gestation than the control, and may reflect the differences between in Treg frequency between PE and controls. This is not specified in the m&m.

Thank you very much for this comment, it is definitely necessary to consider this when interpreting our results.

However, it is shown in the literature that the number of Treg peaks in the second trimester and decreases in the further pregnancy until birth (Figueiredo & Schumacher, 2016; Hosseini, Dolati, Hashemi, Abdollahpour-Alitappeh, & Yousefi, 2018; Robertson et al., 2019; Teles, Zenclussen, & Schumacher, 2013). Therefore, if one assumes the significant difference of Treg cell number in the two groups is because of the differing pregnancy weeks contradictory results would be expected. However, this is definitely an important aspect to discuss in our manuscript, therefore we added corresponding content in our discussion. Further we were able to show in the regression analysis which we performed, that the week of gestation had no significant influence on the results of our investigations, neither on the number of Tregs, nor on the expression of CCL22. Please refer to corresponding graphs in the adjusted supplement.

  1. Fig.5 : Apoptosis of Treg is not convincing at all: only one cell shown in panel A-H!!, no quantification, this is just not scientifically acceptable

Thank you very much for this important annotation We added our quantitative analysis in Fig. 5. The representative images show only few cells since this was representative for our analysis of one visual field. Due to the greatly reduced number of Tregs in the preeclampsia placentas compared to the controls, it was not possible to display more than one Treg on one image which shows still recognizable Tregs. The number of decidual Tregs in the third trimenon is as low as shown in Fig. 1. Analysis was performed with a 40x objective in one visual field. For quantification all cells were counted in one sample. Showing more Tregs in one visual field would only be possible if a disproportionately small magnification was selected, in which, however, double staining would no longer be easily recognizable. When counting all Tregs in the decidua which was shown in each sample there were not more than 20 cells in the control placentas, while in the PE placentas only 11 Treg could be found maximally (Fig. 5). In addition, this is also represented in the mean count of Tregs per visual field shown in the immunohistochemistry (2.1). However, in the images E-H more than one single cell can be detected, but since these cells are not Treg, they are only stained with DAPI in blue (this is a cell nucleus staining) or partly additionally by TUNEL in green. Furthermore, we have added further pictures for you, in a wider field of magnification. If wished, we can add these pictures in our supplement e.g..

Staining colors: blue = DAPI for nucleus staining, red = FoxP3, green = TUNEL

Example of a wider field of magnification in a control placenta. As you can see the smaller the magnification, the harder it gets to distinct a Treg from a FoxP3 stained point that is not considered to be a cell (no DAPI-Staining).

Staining colors: blue = DAPI for nucleus staining, red = FoxP3, green = TUNEL

Example of a wider field of magnification in a PE placenta. Caused by the small amount of Tregs found in the PE placentas in a big amount of fields of views, no Tregs could be detected at all. Therefore, the single Treg represented in the pictures E-H in Figure 5 are not caused by a non-scientific way of work but through the low density of Tregs in the PE placentas.

Staining colors: blue = DAPI for nucleus staining, red = FoxP3, green = TUNEL

Example of a PE placenta with an even higher magnification than shown in the manuscript. Here you can see one apoptotic Treg as well as another apoptotic cell that isn’t FoxP3 positive. We think this is a good picture to distinct between the different stained cells, but it does not provide a good overview over the placental tissue.

  1. Correlation with Gal2 and Treg should be shown. How Gal2 is detected and quantified.

Thank you for this note. We are pleased to explain this better in our manuscript and to you. The Gal-2 expression data that we used for the calculation of correlation with Treg were studied earlier in our laboratory and published by Hutter et al. Gal-2 was also immunohistochemically stained and analyzed by IRS in this study.

  1. The plot for CCL22 and Foxp3 correlation should be shown, even if statistically significant (p=0.04) the correlation is very weak (r=0.2) and not convincing.

 We agree with you that the data needs to be shown, thank you for this, we have added the plot for the correlation. Since according to Cohen (1977) a correlation between r = 0.3 and r = 0.5 is considered to be moderate, our correlation of r = 0.25 may be weak, but in a statistical point of view every significant correlation is a relevant correlation even if it is weak. Moreover, the correlation between Treg and CCL22 is already a well-known and variously proven correlation, which is why we assume this correlation is convincing, especially in combination with our other results. We really hope you find our argumentation appropriate and traceable.

Minor concerns:

  1. Fig.1 A wider field should be shown in panel B/C

We provided a Figure in our Supplement with a wider field. The picture as representative visual field in Fig. 1 A shows a 40x magnification as it was used in our analysis. Cells were counted per visual field and this is shown in our representative pictures. When demonstrating the cell number in a lower magnification, cells are not well identifiable as positive Tregs and analysis is limited. Refer to the photo provided. If you find these pictures more representative a more appropriate please let us know and we will adjust the Figure. 

Representative pictures of PE placentas (bottom row) and controls (upper row) of the immunohistochemical staining of FoxP3. The FoxP3 positive Treg are marked with arrows.

  1. Fig.2/3: what about the other CCR4 ligand CCL17?,

You are absolutely right that there are a variety of chemokines and receptors that influence Tregs. We also mention in our discussion that the influence of other chemokines needs to be evaluated in future studies. In our study, we chose to evaluate CCL22 initially because it is one of the best studied Treg-attracting chemokines.

  1. Expression of CKR on placenta Treg, and other Treg chemokines in the placenta?

We agree with you that it would be very interesting to find out more about the different chemokines of Tregs in the placenta. This will be part of our further studies, since we concentrated more on CCL22 as major Treg chemoattractant, the apoptosis of Tregs and the influence of Gal-2 in our study. But we are about to perform an analysis of CCR4 in our collective, results are awaited in the next months.

Reviewer 1:

The authors provided an interesting paper focused on studying  the recruitment  of Treg cells to the decidua of PE affected pregnancies,  their apoptotic status and tested  ( in vitro ) the contribution of  Galectin 2 as a potential anti-apoptotic protein during preeclampsia  

Thank you very much for appreciating our study results and manuscript. We tried to follow your comments and your suggestions.

# Figures:  The cohort includes 32 PE and 34 Control placentas. It is not clear how many placentas/ deciduas were tested and included in statistic analyses.  It is suggested that  the number of samples   ( N) should be included in  the figures that shows statistic data ( figure 1,2,3 )

We apologize that you considered the inclusion of the cohort in the statistical analysis as unclear. For the immunohistochemical analyses, the entire cohort of each of the 32 PE and 34 control placentas were included. We have added the information to the figure legends of Figures 1-3 according to your request.

# 2.7 Correlation of Gal-2 with Tregs: The data presented under this subsession is not supported by figure, table, how it was tested etc. Data should be shown 

We have added a graphic in the supplementary material. The statistical test used to calculate the correlation can be found in 4.7 statistical analysis and was the Spearman-Rho correlation test for non-parametric tests.

# The  lack of information about the source of the Gal-2 is a real concern. Is it recombinant?? expression sytem?? Purity??

It is well known that nanogram quantities of endotoxin can affect the biochemical events of certain cell and can interfere with in vitro experiments. I recommend that  authors exclude the presence of endotoxin in the Gal-2 

Thank you very much for this important comment. We have added the information regarding the company of recombinant Gal-2 in the manuscript. We are also pleased to provide you with the company's datasheet for the purity of recombinant Gal-2. https://www.novusbio.com/products/galectin-2-recombinant-protein_nbp1-48326

We did not perform any endotoxin measurement, but we would not expect a presence of endotoxins in the selected Gal-2, since it was tested by the company and endotoxins would furthermore influence the survival of the Tregs negatively. I hope you find our argumentation sufficient.

In addition to the groups mentioned in the manuscript, we also performed a measurement of caspase-3 after solitary incubation with Gal-2. However, since it was not possible for us to investigate this group for each donor. Due to the low gain of Tregs, per donor it was not possible to perform this for every donor, therefore it was not mentioned in our manuscript so far.

A representative sample showed no difference between the untreated control group and the Tregs incubated with Gal-2 alone.

# Line 197: The author mentioned that"  the lack of significant differences in the .........seems to occur through donor specific difference ". The results of Capase activity is relying on 3 blood samples. I think that there is a place to increase the no. of samples in order to increase the statistic power . 

Since the isolation process of Tregs is quite complex and cell yield is not very high in every case, several studies where Tregs were isolated for cell culture experiments, performed merely technical triplicates, no biological triplicates, or samples were pooled. Although there is no specific source of literature for it, analysis of a wide variety of papers shows that working as a biological triplicate in cell culture is a long-established scientific standard. Furthermore, you are of course correct that a larger sample would result in a greater statistical power. A higher statistical power would only lead to a smaller type II error (beta-error). Your concern, however, refers to the type I error. However, the type I error is determined by setting the significance level. By adding more donors, there will still be big differences between the donors, which would be not easy to be released. Therefore, we thought a standardization would be necessary in every case. We hope we could take your doubts and that you find our argumentation traceable.

# It is not clear if the authors included a control  galectin ( or other protein) as a negative  control  on apoptosis induced by FasL   ( Figure 6 and  2.8).

We are sorry that we have not expressed ourselves clearly enough. We studied three groups: untreated Tregs to serve as controls, Tregs with FasL to quantify the clear induction of apoptosis as well as its level, and the group of Tregs with FasL and Gal-2. We did not examine other galectins in conjunction with FasL since results of older imunnohistochemical evaluations of our group analyzing several other galectins in the placenta, did not show a correlation with the number of Tregs, therefore we felt that this would not have added value to our question. Please let us know whether you find our argumentation acceptable.

# The cohort includes early and late PE with GA ranging from 25-40 weeks. Is there any differences in Treg level , CCL2 and and appototic Treg cells in eraly vs late PE??. The author should  consider and discuss this point .

You are absolutely right, that a comparison of EOP and LOP can offer interesting insights. Since there was no significant difference between EOP and LOP, we had previously excluded this point. We have mentioned that we did not find any difference in our results in our manuscript and added some thoughts about the pregnancy week in our discussion.

Cohen, J. (1977). Statistical power analysis for the behavioral sciences. New York: Academic Press.

Figueiredo, A. S., & Schumacher, A. (2016). The T helper type 17/regulatory T cell paradigm in pregnancy. Immunology, 148(1), 13-21. doi:10.1111/imm.12595

Hosseini, A., Dolati, S., Hashemi, V., Abdollahpour-Alitappeh, M., & Yousefi, M. (2018). Regulatory T and T helper 17 cells: Their roles in preeclampsia. J Cell Physiol, 233(9), 6561-6573. doi:10.1002/jcp.26604

Robertson, S. A., Green, E. S., Care, A. S., Moldenhauer, L. M., Prins, J. R., Hull, M. L., . . . Dekker, G. (2019). Therapeutic Potential of Regulatory T Cells in Preeclampsia-Opportunities and Challenges. Front Immunol, 10, 478. doi:10.3389/fimmu.2019.00478

Teles, A., Zenclussen, A. C., & Schumacher, A. (2013). Regulatory T cells are baby's best friends. Am J Reprod Immunol, 69(4), 331-339. doi:10.1111/aji.12067
